# Determination of Apoptotic Mechanism of Action of Tetrabromobisphenol A and Tetrabromobisphenol S in Human Peripheral Blood Mononuclear Cells: A Comparative Study

**DOI:** 10.3390/molecules27186052

**Published:** 2022-09-16

**Authors:** Anna Barańska, Bożena Bukowska, Jaromir Michałowicz

**Affiliations:** Department of Biophysics of Environmental Pollution, Faculty of Biology and Environmental Protection, University of Lodz, Pomorska Str. 141/143, 90-236 Lodz, Poland

**Keywords:** tetrabromobisphenol A, tetrabromobisphenol S, peripheral blood mononuclear cells, apoptosis, cytosolic calcium ion level, transmembrane mitochondrial potential, caspase activation, PARP-1 cleavage, chromatin condensation

## Abstract

Background: Tetrabromobisphenol A (TBBPA) is the most commonly used brominated flame retardant (BFR) in the industry. TBBPA has been determined in environmental samples, food, tap water, dust as well as outdoor and indoor air and in the human body. Studies have also shown the toxic potential of this substance. In search of a better and less toxic BFR, tetrabromobisphenol S (TBBPS) has been developed in order to replace TBBPA in the industry. There is a lack of data on the toxic effects of TBBPS, while no study has explored apoptotic mechanism of action of TBBPA and TBBPS in human leukocytes. Methods: The cells were separated from leucocyte-platelet buffy coat and were incubated with studied compounds in concentrations ranging from 0.01 to 50 µg/mL for 24 h. In order to explore the apoptotic mechanism of action of tested BFRs, phosphatidylserine externalization at cellular membrane (the number of apoptotic cells), cytosolic calcium ion and transmembrane mitochondrial potential levels, caspase-8, -9 and -3 activation, as well as PARP-1 cleavage, DNA fragmentation and chromatin condensation in PBMCs were determined. Results: TBBPA and TBBPS triggered apoptosis in human PBMCs as they changed all tested parameters in the incubated cells. It was also observed that the mitochondrial pathway was mainly involved in the apoptotic action of studied compounds. Conclusions: It was found that TBBPS, and more strongly TBBPA, triggered apoptosis in human PBMCs. Generally, the mitochondrial pathway was involved in the apoptotic action of tested compounds; nevertheless, TBBPS more strongly than TBBPA caused intrinsic pathway activation.

## 1. Introduction

Brominated flame retardants (BFRs), including brominated bisphenols, are utilized in the production of various everyday products in order to reduce their flammability [1,2].

The synthesis of tetrabromobisphenol A (TBBPA) accounts for 60% of the total global market for BFRs, making it the most widely used flame retardant worldwide [3]. This substance is utilized in the synthesis of polymers (polycarbonates, epoxy resins) and production of electronic and electric equipment, as well as textiles and furniture [4,5].

Due to the widespread use of TBBPA, this substance has been repeatedly found in the environment [6,7,8], dust of residential and office spaces [9], as well as ambient and indoor air [7,10,11].

TBBPA has also been determined in the human organism. Cariou et al. [12] found TBBPA in samples of mothers milk and serum, as well as fetuses serum in the range of concentrations from 0.06 to 37.34 ng/g of fat, while Fujii et al. [13] detected TBBPA in the plasma of Japanese men in an average concentration of 950 pg/g fresh weight. Some research works have documented the occurrence of significant amounts of TBBPA in the human body. For instance, Lankova et al. [14] determined TBBPA in the concentration of up to 16.2 µg/L in samples of women breast milk.

The ubiquitous presence of TBPPA in the environment and the human body prompted the scientists to assess toxic potential of this substance. The conducted studies have shown pro-oxidative, pro-inflammatory, neurotoxic, nephrotoxic, hepatotoxic, genotoxic and endocrine disrupting potential of TBBPA [15,16,17,18,19,20,21,22,23].

In search of a better and less toxic BFR, tetrabromobisphenol S (TBBPS) has been developed in order to potentially replace TBBPA in the industry [24]. TBBPS is commonly utilized in the synthesis of polycarbonates and epoxy resins, as well as in the production of textiles and electronic devices [25,26,27]. Although TBBPS is still less commonly utilized in the industry than TBBPA, it has been found in the environment, food and indoor air [1,27,28,29,30,31]. Recently, TBBPS was determined in an average concentration of 0.593 μg/L of serum samples collected from pregnant Chinese women [32].

Limited data exist on the effect of TBBPS on living organisms. Ding et al. [33] noticed that TBBPS changed the circadian rhythm network in zebrafish and induced developmental changes in zebrafish embryos. These scientists have considered that TBBPS may not have exhibited lower toxicity, in comparison to TBBPA in humans. Liang et al. [34] noticed that TBBPA and TBBPS revealed comparable toxic effects on human embryonic stem cells. They found that tested substances altered the development of neural ectoderm, influenced the growth of axons and the transmission of neurons, as well as disturbed the WNT and AHR signaling pathways.

Apoptosis is a regulated process that allows the removal of old and damaged cells without inflammatory reactions. However, it has been shown that different stimuli, including toxicants, accelerate apoptosis that may contribute to development of various pathological states [35]. Two main apoptotic pathways are involved in apoptosis of lymphocytes. The intrinsic (mitochondrial) cell death pathway is triggered by various apoptotic stimuli, such as genomic instability, cytokine withdrawal or toxicants. Intrinsic cell death signals generally focus on the cell at the outer membrane of mitochondria, which leads to a loss of mitochondrial membrane integrity, and the subsequent induction of apoptotic pathways, in which various proteases, including essential initiator caspase-9 are activated. The major mediators in the intrinsic cell death pathway are Bcl-2 family proteins, which control mitochondrial membrane integrity [36].

On the other hand, death receptors are involved in the initiation of the extrinsic (receptor) death pathway. All death receptors contain a death domain (DD) in their cytoplasmic tail. There are two major DD-containing adaptor proteins implicated in death receptor signaling, such as Fas-associated death domain (FADD) and tumor necrosis factor (TNF) receptor-associated death domain (TRADD). When ligands, including reactive oxygen species (ROS) and xenobiotics, bind to death receptors, the DD mediates the interaction with other DD-containing adaptor proteins, which activates the extrinsic pathway, in which the induction of other important proteins, such as initiator caspase 8 occurs [37].

PBMCs are directly exposed to toxicants entering the human organism. Those cells play a crucial role in the human body as they are involved in antibodies production, killing virus-infected and cancer cells, as well as regulating the immune system response [38].

It was shown that accelerated apoptosis of PBMCs may be associated with adverse changes in the immune system, such as depleted human immunity [39], which, in consequence, may lead to the development of autoimmune diseases (type 1 diabetes, asthma, allergy) and cancer [40,41,42,43]. 

Some research works have revealed that TBBPA can disturb the function of the immune system. Feiterio et al. [44] proved that TBBPA altered the tumor killing function of NK cells and changed cytokines production, such as interferon gamma (IFNɣ), interleukin-1β (IL-1β) and tumor necrosis factor (TNF). In another work, Hall et al. [45] noticed that, in rats, TBBPA caused the downregulation of genes involved in the immune response, which could have led to estrogen-mediated immunosuppression in studied animals.

According to our best knowledge, no research work has been performed to assess the mechanism of apoptotic action of TBBPA in human leucocytes, and only Cato et al. [46] noticed that TBBPA activated caspase-3 and mitogen-activated protein kinases (MAPKs) in human NK cells. Moreover, the effect of TBBPS on apoptosis induction has not been assessed in any cell type. Our earlier research works have revealed that TBBPA and TBBPS caused ROS formation and induced damage to lipids, proteins and DNA in human PBMCs [22,23], while it is well known that ROS induction and DNA damage can trigger apoptotic cell death [47].

Taking the above into consideration, we compared apoptotic potential of TBBPA and TBBPS in human PBMCs and determined the mechanism underlying the action of these substances by assessing changes in phosphatidylserine translocation in cellular membrane, alterations in cytosolic calcium ion and transmembrane mitochondrial potential levels, and caspase-8, -9 and -3 activation. Moreover, the effect of tested compounds on the cleavage of PARP-1, DNA fragmentation and condensation of chromatin was evaluated.

The investigations were conducted in the range of concentrations of tested compounds that referred to their levels detected in humans environmentally or/and occupationally exposed. If no effects were observed at the above-mentioned BFRs levels, higher concentrations of TBBPA and TBBPS were used. 

## 2. Results

### 2.1. Quantitative Analysis of Apoptosis

TBBPA and TBBPS increased PS translocation in peripheral blood mononuclear cells (PBMCs), which was assessed by annexin V-FITC and PI staining. After 24 h of treatment, TBBPA at 5 µg/mL, 25 µg/mL and 50 µg/mL caused a concentration-dependent increase in the number of apoptotic cells. TBBPS induced substantially lower alterations in examined parameters, increasing the number of apoptotic cells only at 50 µg/mL (Figure 1A,B).

### 2.2. Cytosolic Calcium Ions Level

Alteration in the intracellular calcium ion level were determined using Fluo-3/AM, which is hydrolyzed by membrane esterases to Fluo-3. Fluo-3 fluorescence intensity increases with rise of cytosolic Ca^2+^ level. A concentration-dependent increase in cytosolic Ca^2+^ level was noted in PBMCs incubated with TBBPA at 0.1 µg/mL, 1 µg/mL and 5 µg/mL for 24 h. TBBPS at the same concentrations caused smaller changes in the studied parameter (Figure 2).

### 2.3. Transmembrane Mitochondrial Potential

Transmembrane mitochondrial potential (Δ*Ψm*) was evaluated using MitoLite Red CMXRos probe. Alterations in the intensity of marker fluorescence were associated with changes in Δ*Ψm* level. It was revealed that after 24 h of treatment, TBBPA and TBBPS in a concentration-dependent manner reduced Δ*Ψm* in the incubated cells. TBBPA at 1 µg/mL and 5 µg/mL, and, particularly at 25 µg/mL, substantially reduced the tested parameter, while TBBPS at 5 µg/mL and 25 µg/mL induced a smaller depletion of Δ*Ψm* in PBMCs (Figure 3).

### 2.4. Caspase-8, -9 and -3 Activation

Caspase-8 and caspase-9 belong to the initiator enzymes of programmed cell death. A small rise in caspase-8 activation was shown in tested cells treated for 24 h with TBBPA and TBBPS at 5 µg/mL and 25 µg/mL, whereas TBBPS caused slightly higher caspase-8 activation than TBBPA in tested cells (Figure 4A).

After 24 h of treatment, TBBPA and TBBPS at 5 µg/mL and 25 µg/mL caused a substantial increase in caspase-9 activation. It was observed that TBBPA, more strongly (particularly at its highest concentration) than TBBPS, increased caspase-9 activation in tested cells (Figure 4B). Caspase-3 is an executory enzyme of apoptosis. It was noticed that after 24 h of incubation, both studied BFRs, and more strongly TBBPA at 5 µg/mL and 25 µg/mL raised caspase-3 activation in PBMCs (Figure 4C).

Caspase-8, caspase-9 and caspase-3 inhibitors were also used in these experiments. The preincubation of the cells with appropriate caspase inhibitor depleted caspase activity up to control value (data not shown).

### 2.5. PARP-1 Cleavage

PARP-1 is a substrate for caspase-3. The cleavage of PARP-1 and its inactivation helps cells undergoing apoptosis. To determine PARP1 cleavage, a concentration of 25 µg/mL of TBBPA and TBBPS was selected, which caused the strongest caspase-3 activation in PBMCs.

It was found that after 24 h of treatment, TBBPS and more strongly TBBPA at 25 µg/mL induced PARP-1 cleavage in tested cells (Figure 5).

### 2.6. APO-BrdU TUNEL Assay

APO-BrdU TUNEL assay was used for labeling DNA strand breaks along with total cellular DNA to detect apoptotic cells. It was noticed that after 24 h of treatment, TBBPA and less strongly TBBPS at 25 µg/mL induced substantial DNA fragmentation in PBMCs (Figure 6).

### 2.7. Apoptosis Detection by Fluorescence Microscopy

Figure 7 presents representative photomicrographs of PBMCs stained with Hoechst 33324/PI. Before staining, the cells had been incubated for 24 h with DMSO at 0.2% (control), as well as with TBBPA or TBBPS at 5 µg/mL and 50 µg/mL. In control samples, only viable PBMCs were noticed. In probes incubated with TBBPA at 5 µg/mL mainly viable, as well as single early apoptotic cells were observed, while in the probes treated with TBBPS at 5 µg/mL, mainly viable PBMCs were found. Probes treated with TBBPA at 50 µg/mL contained mostly late apoptotic and necrotic cells, whereas the probes incubated with TBBPS at 50 µg/mL consisted of mainly viable, as well as some early apoptotic and late apoptotic PBMCs.

## 3. Discussion

Apoptosis is a regulated process, which enables the removal of unwanted (old, damaged) cells from organisms without inflammatory reactions. It has been shown that various xenobiotics can trigger apoptosis [35], while excessive apoptosis may contribute to the development of infections, heart failure, neurodegenerative diseases or myocardial ischemia.

Peripheral blood mononuclear cells (PBMCs) are key cells of the immune system and are directly exposed to toxicants entering the human body. The accelerated apoptosis of lymphocytes (main population of PBMCs) caused by action of xenobiotics, such as BFRs, may lead to disorders in the immune system function and possibly be involved in cancer development. Additionally, excessive apoptosis of PBMCs can contribute to the development of allergy and asthma [49].

In this research work, apoptotic potential and mechanism of action of TBBPA and TBBPS in human PBMCs was assessed by analyzing phosphatidylserine (PS) translocation in cellular membrane, determining cytosolic calcium ion and transmembrane mitochondrial potential levels, as well as evaluation of caspase-8, -9 and -3 activation, PARP-1 degradation, chromatin condensation and DNA fragmentation.

Our study revealed that tested compounds induced apoptosis of PBMCs but exhibited different apoptotic potential in studied cells. Both examined compounds increased the number of cells showing PS exposure, while TBBPA caused greater alteration in this parameter, when compared with TBBPS. It was observed that TBBPA from 5 µg/mL induced apoptosis of PBMCs, whereas TBBPS, only at 50 µg/mL, triggered apoptotic cell death (Figure 1). Similar findings were achieved by Mokra et al. [50] who assessed the apoptotic potential of debrominated analogs of TBBPA and TBBPS in PBMCs. They observed that bisphenol A (BPA) exhibited stronger apoptotic potential than bisphenol S (BPS) in this cell type. Cho et al. [51] proved that TBBPA (25–200 µM) after 24 h of incubation induced apoptosis in mouse astrocytes and neural stem cells. The authors reported that TBBPA increased ROS formation, which caused mitochondrial dysfunction and ultimately led to apoptosis through c-Jun N-terminal kinase-p53 pathway activation. In another study, Zhang et al. [52] reported a substantial rise in the number of apoptotic human liver L02 cells treated for 48 h with TBBPA in the range from 5 to 40 µM. TBBPA at 20 µM was also shown to induce significant apoptosis of insulinoma RIN-m5F rat cells after 48 h of incubation [53], while at a low concentration of 81 nM, this substance increased the number of apoptotic HepG2 cells by increasing ROS formation, decreasing Δ*Ψm* and activating Ras signaling pathway [54]. Moreover, Wu et al. [55] observed an increased number of apoptotic cells in the heart and brain of embryos and larvae of Zebrafish exposed to TBBPA.

A crucial point in the apoptosis is the increase in cytosolic calcium ions level. Calcium ions released from endoplasmic reticulum into cytosol accumulate in mitochondria, which may lead to the depletion of Δ*Ψm* and, finally, to the induction of apoptosis [56,57,58].

We observed that tested substances increased the level of cytosolic calcium ions and depleted Δ*Ψm* level from 0.1 µg/mL and 1 µg/mL, respectively, while TBBPA induced greater changes than TBBPS (Figure 2 and Figure 3). Ogunbayo et al. [59] noticed that TBBPA at 30 µM raised Ca^2+^ level and reduced Δ*Ψm* in TM4 mouse Sertoli cells. Similarly, Cho et al. [51] reported that TBBPA from 25 µg/mL depleted Δ*Ψm* in mouse astrocytes and neural stem cells, whereas Lu et al. [54] noticed that this substance at 81 nM depleted Δ*Ψm* in HepG2 cells. Moreover, Mokra et al. [50] observed that BPA and BPS were capable of increasing Ca^2+^ level and decreasing Δ*Ψm* in human PBMCs.

The intrinsic pathway, the so-called mitochondrial pathway, is activated as a result of oxidative stress and DNA damage, as well as by increased cytosolic calcium ion level and disturbances of Δ*Ψm* [60,61]. The main component of the intrinsic pathway is the mitochondria. It has been shown that through the reactions involving Bcl-2 family proteins, mitochondrial permeability transition pores (MTP) are formed that release cytochrome c into mitochondrial intermembrane space. Then, cytochrome c together with procaspase-9, ATP and the cytosolic protein Apaf-1 forms an apoptosome in the cytoplasm that activates initiator caspase-9 [61,62,63]. On the other hand, the extrinsic (receptor) pathway may be activated by ROS, which involves the interactions of ligands with transmembrane receptors leading to the activation of initiator caspase-8 [47,62]. Although, the formation of ROS has not been assessed in this study, our previous work showed that TBBPA and TBBPS at low concentrations increased ROS level in human PBMCs [22].

Generally, our study showed that caspase-9 was more strongly activated than caspase-8 in PBMCs exposed to TBBPA and TBBPS at 5 µg/mL and 25 µg/mL. Moreover, it was observed that TBBPA induced a stronger increase (than TBBPS) in caspase-9 activity, while in the case of caspase-8 activation, the results were opposite (Figure 4A,B). The stronger activation of caspase-9 suggests that mitochondrial pathway was mainly implicated in apoptosis induction of PBMCs exposed to TBBPS and, in particular, TBBPA.

It has been proven that both caspase-8 and caspase-9 can activate caspase-3, which is the executioner enzyme of apoptosis.

Obtained results revealed a substantial rise in caspase-3 activity, which was stronger in cells incubated with TBBPA (Figure 4C). A rise in caspase-8, -9, and -3 activities in human PBMCs exposed to debrominated analogs of tested BFRs, such as BPA and BPS was reported by Mokra et al. [50]. In other studies, Szychowski and Wójtowicz [64] observed that TBBPA (100 nM–100 µM, 6 h of incubation) increased caspase-3 activity in mouse hippocampal neuronal cells, while Al-Mousa and Michelangeli [17] showed that TBBPA in the concentrations from 1 to 30 µM activated caspase-3 in SH-SY5Y neuroblastoma cell line. Caspase-3 activation was also reported by Wu et al. [65] in A549 cells (human lung carcinoma cell line) exposed for 48 h to TBBPA in the concentrations range from 8 to 64 µg/mL. Moreover, Jarosiewicz et al. [66] observed apoptotic changes in human erythrocytes exposed to bromophenolic flame retardants. They demonstrated that TBBPS, and more strongly TBBPA, caused PS externalization (increased the number of apoptotic erythrocytes) and induced caspase-3 activation.

Caspase-3 causes PARP-1 cleavage into fragments of 89 kDa and 24 kDa. Fragments of 24 kDa bind to DNA breaks. In contrast, fragments of 89 kDa with attached PAR polymers are moved from nucleus into cytoplasm and interact with apoptosis-inducing factors (AIF), which, as a consequence, leads to the shrinking of the cell nucleus and apoptosis [67].

In order to determine PARP-1 cleavage, a concentration of 25 µg/mL of TBBPA and TBBPS was chosen, which the most strongly activated caspase-3 in tested cells. It was revealed that TBBPA and less strongly TBBPS at 25 µg/mL caused PARP-1 cleavage in human PBMCs (Figure 5). Similarly, Mokra et al. [50] observed PARP-1 fragmentation in human PBMCs treated with BPA and BPS.

TUNEL method allows the detection of apoptotic DNA fragmentation and is one of the most widely used assays to detect programmed cell death [68]. We used this method to confirm that tested compounds were capable of inducing apoptosis in studied cells.

Our study revealed that TBBPS, and more strongly TBBPA at 25 µg/mL, increased the number of TUNEL-positive cells (Figure 6). Park et al. [69] reported that TBBPA at 125 µg/mL (24 h of incubation) caused caspase-3 activation and increased the number of TUNEL-positive HEI-OC1 cells (mouse auditory cell line). Moreover, an increase in TUNEL-positive cells was noticed by Zatecka et al. [70] in mouse spermatozoa, treated with TBBPA at 200 µg/L dissolved in drinking water.

Staining PBMCs with Hoechst33342/PI and using fluorescence microscopy allowed to visualize differences in apoptotic/necrotic changes induced by tested compounds. We noted that TBBPA, particularly at higher concentration of 50 µg/mL induced greater alterations than TBBPS in studied cells. The samples treated with TBBPA contained mainly late apoptotic and necrotic cells, while the probes exposed to TBBPS showed the presence of mostly early apoptotic and viable PBMCs (Figure 7).

It must be underlined that the determination of the mechanism of proapoptotic action of toxicants, such as TBBPA and TBBPS, is crucial for the recognition of cellular targets, which are affected by studied compounds. It must also be taken into account that alterations in some apoptotic parameters, such as calcium ion level or Δ*Ψm* (that usually occur at relatively low toxicant concentrations), are linked to other cellular processes (e.g., cell signaling, energy state of the cell, etc.), and their disturbance may change cell function before apoptosis occurs.

Summing up, this research work represents a mechanistic approach elucidating the effect of tested BFRs on nucleated blood cells, which brings us closer to understanding the action of these substances in the human organism.

## 4. Conclusions

In conclusion, (1) TBBPA and TBBPS exhibited different apoptotic potential in human PBMCs. (2) Tested compounds triggered apoptosis by PS externalization on cellular membrane, increasing cytosolic Ca^2+^ level, decreasing transmembrane mitochondrial potential, activating caspase-8, -9 and -3, as well as increasing PARP-1 cleavage, DNA fragmentation and chromatin condensation. (3) Tested BFRs induced apoptosis mainly by the involvement of mitochondrial pathway; although TBBPA and TBBPS more strongly activated intrinsic and extrinsic mitochondrial pathways, respectively. (4) Similarly, as in our previous studies on prooxidative and genotoxic potential, stronger apoptotic effects were provoked by TBBPA, in comparison to TBBPS.

## 5. Materials and Methods

### 5.1. Chemicals

Tetrabromobisphenol A (99%, 2,2-bis[3,5-dibromo-4-hydroxyphenyl]propane) was bought from LGC Standards (Teddington, UK). Tetrabromobisphenol S (4,4′-sulfonylbis[2,6-dibromo-phenol]) (98.8%) was synthetized in the Institute of Industrial Organic Chemistry in Warsaw (Warsaw, Poland). Caspase-3 fluorimetric assay kit, HBSS solution, pluronic F-127, Hoechst 33342, propidium iodide and valinomycin were bought in Sigma-Aldrich (St. Louis, MO, USA). Perm/Wash Buffer and FITC Annexin V Apoptosis Detection Kit were obtained from Becton Dickinson (Franklin Lakes, NJ, USA). MitoLite Red CMXRos was bought in AAT Bioquest (Sunnyvale, CA, USA). Fluo-3 AM was obtained in PromoCell (Heidelberg, Germany). Caspase-8 fluorimetric assay kit and caspase-9 chromogenic substrate and caspase-9 inhibitor were bought in BioVision (San Francisco, CA, USA). PARP1 (cleaved Asp214) Monoclonal Antibody (HLNC4) and APO-BrdU TUNEL Assay Kit were obtained from Thermo-Fisher (Waltham, MA, USA). Separation medium (LSM) (1.077 g/cm^3^) and RPMI medium with L-glutamine were purchased from Cytogen (Seoul, South Korea). Camptothecin was bought from Pol-Aura (Dywity, Poland). Ionomycin was obtained from Biokom (Janki, Poland). Other chemicals were obtained from Pol-Aura (Dywity, Poland) and Roth (Karlsruhe, Germany).

TBBPS was synthesized in a reaction of bisphenol S bromination with liquid bromide in concentrated acetic acid at 80 °C [71], according to Figure 1:

Crude residue was rinsed with concentrated acetic acid, dissolved in tetrahydrofuran and precipitated out of the solution using hexane. As a result, TBBPS was obtained, which was a colorless solid with melting temperature of 293.3 °C and HPLC purity of 98.8% (internal standardization), efficiency of the reaction was 35%.

Analysis of TBBPS using mass spectrometry - (EI, 70 eV, *m/z*, int[%]): 570(19), 569(9), 568(67), 567(14), 566(100, M), 565(100, 564(69), 562(18), 502(8), 315(14), 302(120, 301(35), 300(23), 299(64), 298(12), 297(33), 270(18), 269(13), 268(38), 267(22), 266(19), 265(11), 253(10), 252(17), 251(25), 250(14), 249(10), 223(10), 221(10), 172(24), 171(16), 170(25), 169(10), 143(14), 141(13), 91(30), 90(27), 63(33), 62(43), 61(13), 53(13).

### 5.2. Methods

#### 5.2.1. Cell Isolation and Treatment

The method of peripheral blood mononuclear cells (PBMCs) isolation was described in detail by Włuka et al. [22]. PBMCs were isolated from the leukocyte-platelet buffy coat, which was separated from whole blood in the Blood Bank in Lodz, Poland. Blood was taken from healthy, non-smoking volunteers (aged 18–45) who showed no signs of infection disease symptoms. The study was approved by Bioethical Commission of Scientific Research at the University of Lodz, no. 1/KBBN-UŁ/II/2017.

Tested compounds were dissolved in DMSO. Final concentration of DMSO in untreated samples (negative control) and samples incubated with TBBPA or TBBPS was 0.2%. The above concentration of DMSO was not toxic for PBMCs, as evaluated by all studied parameters.

The cells were exposed to tested substances in the range of concentrations from 0.01 to 50 µg/mL (depending on the method used) for 24 h at 37 °C in 5% CO_2_ atmosphere in total darkness. The final PBMCs density used in the experiments (after addition of BFR solution) was 4 × 10^6^ cells/mL for caspase analysis and 1 × 10^6^ cells/mL for assessment of other tested parameters. The viability of PBMCs in negative control was over 95%.

The lowest tested BFRs concentration used in this study corresponded to TBBPA concentration determined in humans environmentally exposed to this substance [14]. If no effects were observed at the above-mentioned BFRs levels, higher TBBPA and TBBPS concentrations were used.

#### 5.2.2. Quantitative Determination of Apoptosis (Annexin V-FITC/PI Staining)

During apoptosis phosphatidylserine (PS) is translocated from the inner to the outer leaflet of the plasma membrane. The method is based on the ability of annexin V (labelled with fluorescein isothiocyanate—FITC) to bind to PS located on the outer monolayer of the cellular membrane. Propidium iodide (PI) is used in order to detect necrotic cells as it enters the cell through damaged membrane and binds to DNA. The experiment was conducted according to manufacturer’s procedure, as described by Jarosiewicz et al. [66].

PBMCs were incubated with TBBPA or TBBPS in the range from 1 to 50 µg/mL for 24 h at 37 °C in total darkness. After incubation, the cells were centrifuged (300× *g*) for 5 min at 4 °C and suspended in RPMI medium. Then, PBMCs were stained with the mixture of Annexin V-FITC and PI (1 µM each) dissolved in Annexin V-binding buffer and incubated for 20 min at room temperature in total darkness. In the cells, apoptosis was induced with camptothecin at 10 μM (positive control). The samples were detected by flow cytometry (LSR II, Becton Dickinson, Franklin Lakes, NJ, USA) (excitation/emission maxima: 488/525 nm for annexin V and 530/620 nm for PI, respectively). FMC gate on PBMCs was established and the data were recorded for a total of 10,000 cells per sample.

#### 5.2.3. Cytosolic Calcium Ion Level

One of the primary parameter of apoptosis is the rise in the level of cytosolic Ca^2+^, which can be measured using stain Fluo-3/AM. Fluo-3/AM shows very low fluorescence, but after its hydrolysis by membrane esterases (Fluo-3 formation) and complexation with calcium ions, it exhibits about 100-fold increase in green fluorescence intensity. The analysis of calcium ion level in human PBMCs was previously described in detail by Mokra et al. and Barańska et al. [50,72].

The cells were incubated with TBBPA or TBBPS in the range from 0.01 to 5 µg/mL for 24 h at 37 °C in total darkness. Then, PBMCs were centrifuged (300× *g*) for 5 min at 4 °C, resuspended in solution of Fluo-3 AM (1 µM), and incubated for 20 min at 37 °C in total darkness. In the next step, HBSS consisting of 1% BSA was added to the cells, which were incubated for 40 min at 37 °C in total darkness. PBMCs were rinsed twice using HEPES buffer, and then centrifuged (300× *g*) for 5 min at 4 °C. Finally, the cells were resuspended in HEPES buffer, and incubated for 10 min at 37 °C in total darkness. Positive control contained the cells treated with ionomycin at 1 µM (calcium ionophore). The samples were detected using flow cytometry (LSR II, Becton Dickinson, Franklin Lakes, NJ, USA) (excitation/emission maxima: 488/525 nm for Fluo-3). FMC gate on PBMCs was established, and the data were recorded for a total of 10,000 cells per sample.

#### 5.2.4. Mitochondrial Transmembrane Potential

A characteristic sign of early apoptosis is a reduction in transmembrane mitochondrial potential (Δ*Ψm*), which can be determined by changes in the level of fluorescence of MitoLite Red CMXRos (excitation/emission maxima: 579/599 nm). This fluorescent stain is a cationic dye that readily penetrates living cells and accumulates in mitochondria, depending on the value of Δ*Ψm*. Due to the presence of thiol-reactive chloromethyl moieties, this stain is retained in the mitochondria [50].

PBMCs were treated with TBBPA or TBBPS in the range of the concentrations from 0.01 to 25 µg/mL for 24 h at 37 °C in total darkness. Nigericin and valinomycin (1 µM), which are capable of increasing and decreasing Δ*Ψm*, respectively, were used as positive controls. After the probes had been incubated, they were centrifuged (300× *g*) for 5 min at 4 °C. The supernatant was discarded, and the cells were resuspended in PBS solution. PBMCs were stained with MitoLite CMXRos at 1 µM, and then incubated for 20 min at 37 °C in total darkness. The probes were determined in 96-well plates using a microplate reader (Cary Eclipse, Varian, Waltham, MA, USA).

#### 5.2.5. Caspase-3, -8, -9 Activation

The process of apoptosis is regulated directly and indirectly by caspases. Fluorimetric analysis of caspase-3 and caspase-8 activation was conducted according to the manufacturers’ protocols with slight modifications [72].

The methods of caspase-3 and caspase-8 detection are based on the hydrolysis of peptide substrates acetyl-Asp-Glu-Val-Asp-7-amino-4-methylcoumarin (Ac-DEVD-AMC) and acetyl-Ile-GluThr-Asp-7-amino-4-methylcoumarin (Ac-IETD-AMC), respectively. Hydrolysis of the substrates results in the release of fluorescent 7-amino-4-methylcoumarin (AMC) (excitation/emission maxima: 360/460 nm). Colorimetric determination of caspase-9 activity was associated with hydrolysis of the substrate Acetyl-Leu-Glu-His-Asp-p-nitroaniline (Ac-LEHD-pNA), which resulted in a release of p-nitroaniline (pNA) (absorption at 405 nm). In all experiments, positive controls were employed that contained cells suspension treated with camptothecin (10 µM). Preincubation with caspases inhibitors was also executed for all experiments. Determination of caspase-3 and caspase-8 activity was performed using a fluorescent microplate reader (Fluoroskan Ascent FL, Labsystem, Thermo-Fisher, Waltham, MA, USA), whereas detection of caspase-9 activity was done using an absorbance microplate reader (BioTek ELx808, Bio-Tek, Santa Clara, CA, USA).

#### 5.2.6. PARP-1 Cleavage

Human poly (ADP-ribose) polymerase (PARP1) is a 116 kDa nuclear enzyme implicated in DNA repair. During apoptosis, caspase-3 cleaves PARP1 between Asp214 and Gly215, generating two fragments of 85 kDa and 25 kDa. The HLNC4 antibody (conjugated with Alexa fluor 488) added to the sample specifically recognizes the 85 kDa PARP1 fragment formed after the enzyme cleavage.

PBMCs were incubated with TBBPA or TBBPS at 25 ug/mL for 24 h at 37 °C in total darkness. Then, PBMCs were washed and suspended in 1% paraformaldehyde (dissolved in PBS). Then, HLNC4 antibody conjugated with Alexa fluor 488 was added to the probes, which were incubated for 30 min at 37 °C in total darkness. In the cells, apoptosis was induced by camptothecin at 10 μM (positive control). Cytometric analysis of the samples was conducted (LSR II, Becton Dickinson, Franklin Lakes, NJ, USA) at excitation/emission maxima of 494/519 nm for Alexa Fluor 488. FMC gate on PBMCs was established for data acquisition, and the data were recorded for 10,000 cells per sample.

#### 5.2.7. APO-BrdU TUNEL Assay

The TUNEL method detects apoptosis by labelling the 3′-OH ends of single- and double-stranded DNA fragments with labelled Br-dUTP (brominated deoxyuridine triphosphate nucleotides). The reaction is catalyzed by terminal deoxynucleotidyl transferase (TdT) [73]. 

The cells were incubated with TBBPA or TBBPS at 25 µg/mL for 24 h at 37 °C in total darkness. Then, PBMCs were fixed in 1% paraformaldehyde. The samples were incubated (1 h at 37 °C in total darkness) in DNA labelling solution containing BrdUTP and TdT. Then, anti BrdUTP antibody was added to the probes, which were incubated for 30 min at room temperature in total darkness. In the cells, apoptosis was induced by camptothecin at 10 μM (positive control). The probes were determined by flow cytometry (LSR II, Becton Dickinson, Franklin Lakes, NJ, USA). Maxima of excitation and emission for Alexa Fluor 488 were 494 nm and 519 nm, respectively. FMC gate on PBMCs was established for data acquisition, and the data were recorded for 10,000 cells per sample.

#### 5.2.8. Apoptosis Determination by Fluorescence Microscopy

Apoptotic changes in tested cells incubated with TBBPA or TBBPS were evaluated by their staining with Hoechst 33342 and PI and analyzing by fluorescence microscopy. The method was described by Rogalska et al. [48], while preparation of the samples (PBMCs) was presented in detail by Mokra et al. and Barańska et al. [50,72].

#### 5.2.9. Statistical Analysis

Data are shown as average values with standard deviation. ANOVA (one-way analysis of variance) test and Tukey’s post-hoc test or Welch’s test (analysis of PARP1 cleavage and DNA fragmentation) were employed in order to evaluate statistical significance between examined probes [74]. Statistical significance was *p* < 0.05. All analyses were done using STATISTICA 13 software (StatSoft, Inc., Tulusa, OK, USA). The experiments were done on blood taken from 4 donors, while for each individual experiment (one blood donor), an experimental point was a mean value of 2–3 replications.

## Data Availability

The raw data supporting the conclusions of this paper are deposited in the Department of Biophysics of Environmental Pollution, University of Lodz and will be made available by the authors, without undue reservation.

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
