# Peer review of "Determination of Apoptotic Mechanism of Action of Tetrabromobisphenol A and Tetrabromobisphenol S in Human Peripheral Blood Mononuclear Cells: A Comparative Study"

_molecules, 2022, doi:10.3390/molecules27186052_

Round 1

Reviewer 1 Report

This paper fits the scope of the journal. The manuscript compares the apoptotic mechanism of TBBPA and TBBPS in PBMCs.

The manuscript is well written, and the results are presented in a clear style and in a logical way.

Still there are some minor corrections that need to be addressed before publication.

Minor points: 

1.     Title : “cells, a comparative study.” Should be replaced by “A comparative study on the…” or “cells: a comparative study.”

2.     Line 11: Please change “BFRs” with “BFR”.

3.     Line 12: Please change “, and the human body” with “and in the human body”;

4.     Line 61: Please change “Chinese” with “chinese”.

5.     Line 109: Please change “as assessed” with “was assessed”.

6.     The legends of figures don’t mention the number of experiments made. Please include this information. Additionally, in some legends there are different font size. Please, correct this.

7.     The data presented in Figure 6 has a huge error. How can you conclude if TBBPA or TBBPS induces more alterations in DNA fragmentation?

8.     TBBPS was prepared in the Institute of Industrial Organic Chemistry in Warsaw. Please provide the methodology for TBBPS synthesis, including its characterization data (NMR, MS, HPLC purity).  

Author Response

Reviewer 1

This paper fits the scope of the journal. The manuscript compares the apoptotic mechanism of TBBPA and TBBPS in PBMCs. The manuscript is well written, and the results are presented in a clear style and in a logical way. Still there are some minor corrections that need to be addressed before publication.

Dear Reviewer,

Thank you very much for favorable evaluation of our manuscript and your valuable comments:

Minor points: 

  1. Title : “cells, a comparative study.” Should be replaced by “A comparative study on the…” or “cells: a comparative study.” 

The title of the paper was corrected

  1. Line 11: Please change “BFRs” with “BFR”.

‘BFRs’ were changed to ‘BFR’

  1. Line 12: Please change “, and the human body” with “and in the human body”;

This sentence was corrected

  1. Line 61: Please change “Chinese” with “chinese”.

This phrase was corrected

  1. Line 109: Please change “as assessed” with “was assessed”.

This phrase was corrected

  1. The legends of figures don’t mention the number of experiments made. Please include this information. Additionally, in some legends there are different font size. Please, correct this.

The number of experiments (the number of blood donors) was added to figures’ legends according to reviewer comment. Font size in all legends was unified.

  1. The data presented in Figure 6 has a huge error. How can you conclude if TBBPA or TBBPS induces more alterations in DNA fragmentation?

In order to demonstrate statistically significant differences both between control and tested compounds and between the compounds themselves, we performed a detailed statistical analysis. Our results were converted to % relative to the control, and were therefore treated as unpaired samples in the statistical analysis. The first step during the analysis was to check normality with the Shapiro-Wilk test and homogeneity of variance with the Brown-Forsythe test. All analyses showed homogeneity of variance and for this reason we finally used for data (contained in figure 6) the Welch test, which confirmed statistically significant differences between control vs. TBBPA, control vs. TBBPS and TBBPA vs. TBBPS.

  1. TBBPS was prepared in the Institute of Industrial Organic Chemistry in Warsaw. Please provide the methodology for TBBPS synthesis, including its characterization data (NMR, MS, HPLC purity).  

According to reviewer comment, the methodology for TBBPS synthesis and characterization was included in section: 5.1. Chemicals. The following information was added:

TBBPS was synthesized in a reaction of bisphenol S bromination with liquid bromide in concentrated acetic acid at 80 °C [Lue et al. 2010] according to the scheme:

Crude residue was rinsed with concentrated acetic acid, dissolved in tetrahydrofuran, and precipitated out of the solution using hexane. As a result, TBBPS was obtained, which was a colorless solid with melting temperature of 293.3 °C and HPLC purity of 98.8% (internal standardization), efficiency of the reaction was 35%.

Analysis of TBBPS using mass spectrometry - (EI, 70 eV, m/z, int[%]): 570(19), 569(9), 568(67), 567(14), 566(100, M), 565(100, 564(69), 562(18), 502(8), 315(14), 302(120, 301(35), 300(23), 299(64), 298(12), 297(33), 270(18), 269(13), 268(38), 267(22), 266(19), 265(11), 253(10), 252(17), 251(25), 250(14), 249(10), 223(10), 221(10), 172(24), 171(16), 170(25), 169(10), 143(14), 141(13), 91(30), 90(27), 63(33), 62(43), 61(13), 53(13).

Reviewer 2 Report

The paper studies the effects of Tetrabromobisphenol A (TBBPA) and Tetrabromobisphenol S (TBBPS) in human peripheral blood mononuclear cells (PBMCs) in terms of induction of apoptosis. The work is interesting, however, before it can be considered for publication, the following issues should be addressed.

The authors provide the full name of acronym, PBMCs, only in the abstract or introduction.

The Introduction section gives an idea of what hypotheses the authors want to address. However, more details should be given on apoptosis process (e.g., two pathways are involved in apoptosis that work synergistically to assure the removal of the defective cells. The intrinsic cell death pathway, or mitochondrial pathway, is activated by the cell itself upon detection of cell damage via a number of intracellular sensors and the extrinsic cell death pathway is activated by the interaction between a cell of the immune system and a damaged cell, and so on….). I think that can help readers understand more clearly the effect induced by tested compounds.

The methods are adequately addressed but I suggest, to provided more information on assays used to tested the involvement of apoptosis process although many references are written. Have the authors considered positive controls for each test? Please, the authors provide this information.

In the results section:

For each figure, the authors indicate the number of experiments.

Please, check the error bars of your manuscript regarding Figure 2 (0.1 µg/mL), Figure 3 (1, 5, 25 µg/mL), Figure 4 B and C (5 µg/mL).

In Figure1, panel B, the control sample show a % Q2+Q4 of 0.2, on the contrary in panel A, control samples are about 10% and SD is small, I do not understand. The authors provide a rationale.

In Figure 6 (25 µg/mL) the standard deviation is very high, why? How many experiments were performed? The authors considered the positive control?

In Figure 7, the authors provide the scale bar to images.

Author Response

Reviewer 2

The paper studies the effects of Tetrabromobisphenol A (TBBPA) and Tetrabromobisphenol S (TBBPS) in human peripheral blood mononuclear cells (PBMCs) in terms of induction of apoptosis. The work is interesting, however, before it can be considered for publication, the following issues should be addressed.

Dear Reviewer,

Thank you very much for favorable evaluation of our manuscript and your valuable comments:

The authors provide the full name of acronym, PBMCs, only in the abstract or introduction.

Full name of PBMCs were added into ‘Materials and methods’, ‘Results’ and ‘Discussion’ sections

The Introduction section gives an idea of what hypotheses the authors want to address. However, more details should be given on apoptosis process (e.g., two pathways are involved in apoptosis that work synergistically to assure the removal of the defective cells. The intrinsic cell death pathway, or mitochondrial pathway, is activated by the cell itself upon detection of cell damage via a number of intracellular sensors and the extrinsic cell death pathway is activated by the interaction between a cell of the immune system and a damaged cell, and so on….). I think that can help readers understand more clearly the effect induced by tested compounds. 

According to reviewer comment, the following description on apoptosis pathways was added into ‘Introduction’

Two main apoptotic pathways are involved in apoptosis of lymphocytes. The intrinsic (mitochondrial) cell death pathway is triggered by various apoptotic stimuli, such as genomic instability, cytokine withdrawal or toxicants. Intrinsic cell death signals generally focus on the cell at the outer membrane of mitochondria, which leads to a loss of mitochondrial membrane integrity, and the subsequent induction of apoptotic pathways, in which various proteases, including essential initiator caspase-9 are activated. The major mediators in the intrinsic cell death pathway are Bcl-2 family proteins, which control mitochondrial membrane integrity (Zhang et al. 2005).

On the other hand, death receptors are crucial in the initiation of the extrinsic (receptor) death pathway. All death receptors contain a death domain (DD) in their cytoplasmic tail. There are two major DD-containing adaptor proteins implicated in death receptor signaling, such as Fas associated death domain (FADD) and tumor necrosis factor (TNF) receptor associated death domain (TRADD). When ligands, including ROS and xenobiotics bind to death receptors, the DD mediates interaction with other DD-containing adaptor proteins, which activates the extrinsic pathway, in which the induction of other important protein, such as initiator caspase 8 occurs (Karuma et al. 2008).

The methods are adequately addressed but I suggest, to provided more information on assays used to tested the involvement of apoptosis process although many references are written. Have the authors considered positive controls for each test? Please, the authors provide this information.

According to reviewer comments we provided more detailed information on methods concerning quantitative detection of apoptosis (Annexin V-FITC/PI staining), as well as calcium ion level and caspases determination.

5.2.2 Quantitative determination of apoptosis (Annexin V-FITC/Propidium iodide staining)

PBMCs were incubated with TBBPA and TBBPS in the range from 1 to 50 µg/mL for 24 h at 37 ºC in total darkness. After incubation, the cells were centrifuged (300g) for 5 min at 4 ºC and suspended in RPMI medium. Then, PBMCs were stained with the mixture of Annexin V-FITC and PI (1 µM each) dissolved in Annexin V-binding buffer, and incubated for 20 min at room temperature in total darkness. In the cells, apoptosis was induced with camptothecin at 10 μM (positive control). The samples were detected by flow cytometry (LSR II, Becton Dickinson) (excitation/emission maxima: 488/525 nm for annexin V and 530/620 nm for PI, respectively). FMC gate on cells was established and the data was recorded for a total of 10,000 cells per sample.

5.2.3 Cytosolic calcium ion level

The cells were incubated with TBBPA and TBBPS in the range from 0.01 to 5 µg/mL for 24 h at 37 ºC in total darkness. Then, PBMCs were centrifuged (300g) for 5 min at 4 ºC, resuspended in solution of Fluo-3 AM (1 µM), and incubated for 20 min at 37 ºC in total darkness. In the next step, HBSS consisting of 1% BSA was added to the cells, which were incubated for 40 min at 37 ºC in total darkness. PBMCs were rinsed twice using HEPES buffer, and then centrifuged (300g) for 5 min at 4 ºC. Finally, the cells were resuspended in HEPES buffer, and incubated for 10 min at 37 ºC in total darkness. Positive control contained PBMCs treated to ionomycin at 1 µM (calcium ionophore). The samples were detected using flow cytometry (LSR II, Becton Dickinson) (excitation/emission maxima: 488/525 nm for fluo-3). FMC gate on PBMCs was established, and the data was recorded for a total of 10,000 cells per sample.

5.2.5 Caspase-3, -8, -9 activation

The process of apoptosis is regulated directly and indirectly by caspases. Fluorimetric analysis of caspase-3 and caspase-8 activation was conducted according to the manufacturers’ protocols with slight modifications [48].

The methods of caspase-3 and caspase-8 detection are based on the hydrolysis of peptide substrates acetyl-Asp-Glu-Val-Asp-7-amino-4-methylcoumarin (Ac-DEVD-AMC) and acetyl-Ile-GluThr-Asp-7-amino-4-methylcoumarin (Ac-IETD-AMC), respectively. Hydrolysis of the substrates results in the release of fluorescent 7-amino-4-methylcoumarin (AMC) (excitation/emission maxima: 360/460 nm). Colorimetric determination of caspase-9 activity was associated with hydrolysis of the substrate Acetyl-Leu-Glu-His-Asp-p-nitroaniline (Ac-LEHD-pNA), which resulted in a release of p-nitroaniline (pNA) (absorption at 405 nm). In all experiments positive controls were employed that contained cells suspension treated with camptothecin (10 µM). Preincubation with caspases inhibitors was also executed for all experiments. Determination of caspase-3 and caspase-8 activity was performed using a fluorescent microplate reader (Fluoroskan Ascent FL, Labsystem), whereas detection of caspase-9 activity was done using an absorbance microplate reader (BioTek ELx808, Bio-Tek).

For analysis of each parameter one positive control was used, while during analysis of transmembrane mitochondrial potential two positive controls were employed. These information is contained in the manuscript.

For each figure, the authors indicate the number of experiments.

The number of experiments (the number of blood donors) was added to figures’ legends.

Please, check the error bars of your manuscript regarding Figure 2 (0.1 µg/mL), Figure 3 (1, 5, 25 µg/mL), Figure 4 B and C (5 µg/mL).

We checked error bars. W tried to insert error bars once again, but we were not able to correct this slight graphic element.

In Figure1, panel B, the control sample show a % Q2+Q4 of 0.2on the contrary in panel A, control samples are about 10% and SD is small, I do not understand. The authors provide a rationale.

Thank you very much for this comment. In Figure 1B, we made a mistake by inserting a dot-plot with PBMCs (stained with Annexin V-FITC/PI) after their isolation (before 24 h of incubation) to check the percent of viable and apoptotic cells at the beginning of the experiment. The figure 1B was corrected, and now the control dot-plot represents PBMCs, which were incubated with 0.2% DMSO for 24 h (negative control).

In Figure 6 (25 µg/mL) the standard deviation is very high, why? How many experiments were performed? The authors considered the positive control?

The positive control included the cells incubated with camptothecin
at 10 µg/mL.

We conducted 3 experiments (blood from 3 donors was used), while an experimental point was a mean value of 3 replications. It is difficult to answer the question, why there is such high standard deviation value. It is probable that higher number of the experiments (analysis of higher number of samples from blood donors) would provide lower value of standard deviation. Unfortunately, there was limitation in financial resources that were intended for grant financing.

It is important to note that in order to demonstrate statistically significant differences both between control and tested compounds and between the compounds themselves, we performed a detailed statistical analysis. Our results were converted to % relative to the control, and were therefore treated as unpaired samples in the statistical analysis. The first step during the analysis was to check normality with the Shapiro-Wilk test and homogeneity of variance with the Brown-Forsythe test. All analyses showed homogeneity of variance and for this reason we finally used for data (contained in figure 6) the Welch test, which showed statistically significant differences between control vs. TBBPA,
control vs. TBBPS, as well as TBBPA vs. TBBPS.

In Figure 7, the authors provide the scale bar to images. 

The scale bar was added to figure 7.
